# Energy dissipation from a correlated system driven out of equilibrium

J.D. Rameau[1], S. Freutel[2], A.F. Kemper[3,4], M.A. Sentef[5,6], J.K. Freericks[7], I. Avigo[2], M. Ligges[2], L. Rettig[2,†], Y. Yoshida[8], H. Eisaki[8], J. Schneeloch[1], R.D. Zhong[1], Z.J. Xu[1], G.D. Gu[1], P.D. Johnson[1] & U. Bovensiepen[2]

In complex materials various interactions have important roles in determining electronic properties. Angle-resolved photoelectron spectroscopy (ARPES) is used to study these processes by resolving the complex single-particle self-energy and quantifying how quantum interactions modify bare electronic states. However, ambiguities in the measurement of the real part of the self-energy and an intrinsic inability to disentangle various contributions to the imaginary part of the self-energy can leave the implications of such measurements open to debate. Here we employ a combined theoretical and experimental treatment of femtosecond time-resolved ARPES (tr-ARPES) show how population dynamics measured using tr-ARPES can be used to separate electron–boson interactions from electron–electron interactions. We demonstrate a quantitative analysis of a well-defined electron–boson interaction in the unoccupied spectrum of the cuprate $Bi_2Sr_2CaCu_2O_{8+x}$ characterized by an excited population decay time that maps directly to a discrete component of the equilibrium self-energy not readily isolated by static ARPES experiments.

[1] Brookhaven National Laboratory, Condensed Matter Physics and Materials Science Department, Brookhaven National Laboratory, 734 Brookhaven Avenue, Upton, New York, 11973, USA. [2] Faculty of Physics and Center for Nanointegration Duisburg-Essen (Cenide), University Duisburg-Essen, Lotharstrasse 1, Duisburg 47057, Germany. [3] Department of Physics, North Carolina State University, Raleigh, North Carolina 27695, USA. [4] Lawrence Berkeley National Laboratory, 1 Cyclotron Road, Berkeley, California 94720, USA. [5] HISKP, University of Bonn, Bonn 53115, Germany. [6] Max Planck Institute for the Structure and Dynamics of Matter, Center for Free Electron Laser Science, Hamburg 22761, Germany. [7] Department of Physics, Georgetown University, Washington, District of Columbia 20057, USA. [8] National Institute of Advanced Industrial Science and Technology, Tsukuba, Ibaraki 305-8568, Japan. † Present address: Fritz-Haber-Institut der Max-Planck-Gesellschaft, Faradayweg 4-6, Berlin 14195, Germany. Correspondence and requests for materials should be addressed to J.D.R (email: jrameau@bnl.gov) or to A.F.K. (email: akemper@ncsu.edu).

A host of interactions have a role in determining the properties of complex quantum materials. Foremost among these are interactions between the electron quasiparticles, and between electron quasiparticles and bosonic excitations such as phonons and spin fluctuations[1]. Quantifying these interactions is essential to understanding the fundamental and emergent phenomena in complex materials. In particular, in strongly correlated materials the phenomenology of these interactions remains vital to understanding the behavior of charge and spin density waves, electrical conductivity anomalies and unconventional superconductivity. However, the analysis is challenging because different experimental methods can easily lead to different conclusions. The difficulty primarily arises because the interactions overlap in energy, making them difficult to disentangle.

Interactions quite generally give rise to a finite quasiparticle lifetime, or equivalently an excitation energy linewidth. These lifetimes are described by the energy-dependent imaginary part of the quasiparticle self-energy, $Im\Sigma(E)$, which represents the effect of the interactions on the quasiparticle at energy $E$ and determines various properties of the material. Each of the interactions has a specific contribution to $Im\Sigma(E)$, which in equilibrium add to result in a total linewidth according to Matthiessen's rule. This superposition of single-particle self-energies complicates the disentanglement of the individual interactions, which is desirable to be able to understand the electron–boson (e–b) and electron–electron (e–e) interactions.

Quasiparticle lifetimes and their related mass renormalizations can in some cases be directly measured using angle-resolved photoemission spectroscopy (ARPES) by studying the linewidth $\Gamma(E)$ or effective dispersion $E(k)$, obtaining the imaginary or real parts of the self-energy ($Im\Sigma(E)$ and $Re\Sigma(E)$), respectively[1–6].

Recently, femtosecond time-resolved ARPES (tr-ARPES)[7–12] has come to the forefront as another method for studying the quasiparticle lifetimes through analyzing population dynamics, and thus ostensibly the interactions. In these experiments, an ultrafast laser pump excites quasiparticles from their ground state, and their subsequent energy- and momentum-resolved relaxation to equilibrium is probed by analysis of the photoelectron spectrum generated by a second, much weaker, time-delayed probe pulse in the UV or extreme XUV spectral range. The lifetime analysis of the laser-excited population relaxation by kinetic rate equations or density matrix formalism[7,13] has proven successful for quasiparticle energies between $\sim 0.5$ to several eV in a limit where relaxation is dominant at all times and quasiparticle lifetimes can indeed be determined in the time domain[7,14–16].

At energies close to the Fermi level $E_F$, the situation is more challenging because two effects become important and hamper the lifetime determination. Immediately after the pump, the primary electronic excitations relax by e–e scattering, which leads to a population redistribution of secondary electronic excitations. Subsequently, coupling to phonons becomes in general the dominant contribution[5,8]. Under the generated non-equilibrium conditions, the various interactions do not simply add to determine the population dynamics and a quantitative, mode-selective analysis was so far not obtained[17].

Here the population dynamics are calculated using a numerical time-dependent Green's function approach[18]. Explicit inclusion of the pump light field in our theoretical description as the primary electronic excitation mechanism avoids the assumption of an initially thermalized electronic distribution which was required in previous work[8] and in the often-used $n$-temperature models. This shows the photoexcited electrons scatter on an ultrashort timescale by e–e interactions and, because these interactions conserve the total energy in the electron system, they cannot be responsible for the dissipation of the excess electronic energy. The dissipation required to relax the system is achieved by electron–phonon (e–p) interactions which transfer the energy from the electron system to a heat bath consisting of $\hbar\Omega \approx 75$ meV phonons, where $\Omega$ is the phonon frequency, which will subsequently decay through anharmonic coupling into acoustic phonons. We validate this assertion of dissipation-driven population dynamics by a comparison between the tr-ARPES measurements and a theoretical calculation which only includes mass renormalizations in the relaxation dynamics.

A measurement of population dynamics on the copper oxide material Bi2212 in its normal state is used both because this material is known to host very strong mass renormalizations throughout its phase diagram[1,2,19] and to demonstrate the ability of our approach to discern such interactions even when obscured by by additional phenomena, as is the case in the normal state of Bi2212. It is the fact that the same e–b coupling contributes to the linewidth of equilibrium ARPES and determines the energy dissipation rate of populations in tr-ARPES that makes studying its properties important, especially since they can be isolated in the time domain.

Here we show how to separate the microscopic interactions of interest using population dynamics observed in tr-ARPES. The experimentally observed relaxation dynamics are found to exhibit three characteristic timescales, $\tau_{ee}$, $\tau_{short}(E)$ and $\tau_{long}(E)$, reflected in the initial pump-induced, energy-dependent peak intensity and the shorter and longer exponential decay constants, respectively. These timescales are assigned, respectively, to redistribution of the optically excited primary electrons, energy dissipation by a single bosonic mode and further relaxation by a bath of acoustic modes. This tripartite enumeration of timescales represents an important step beyond previous tr-ARPES experiments, which have identified only one timescale[17]. The newly discovered short timescale $\tau_{short}(E)$ exhibits a sudden speedup at energies above $+\hbar\Omega$ and is found to account for a large part of the lifetime obtained from the equilibrium linewidth $\tau_{QP}(E)$ measured near $E_F$ along the zone diagonal of Bi2212. At the same time, $\tau_{ee}$ is found by fitting to be $< 10$ fs, which is considerably shorter than $\tau_{short}(E)$, and $\tau_{long}(E)$ is found to consistently be an order of magnitude longer than $\tau_{short}(E)$. These findings open the way to a quantitative understanding of ARPES self-energies through population lifetimes measured in tr-ARPES. By comparison with theory, we show that this timescale can arise from the coupling to a single optical phonon mode[2,20], suggesting a new use for time-resolved measurements—as a selective probe of e–p interactions in materials where their presence and strength are obscured by other processes. This will furthermore allow the separation of e–e from e–p interactions in the single-particle spectrum both in and out of equilibrium. Employing non-equilibrium Green's functions theory, we demonstrate how to make quantitative conclusions based on these observations, and show also that the e–p coupling leaves its mode-specific fingerprint on the population dynamics. Using tr-ARPES, we therefore observe population relaxation dynamics, which we describe by interaction with a specific phonon after excitation by a light field. We find quantitative agreement between experiment and theory.

## Results

**Theory**. We begin with some general remarks regarding the dynamics of population distributions when driven out of equilibrium. The pump deposits energy into the electrons. If e–e interactions are the only interactions present, the excess energy must remain in the electronic system, and although the scattering can cause a rapid change in the energy and momentum

dependent population, it may only thermalize the electrons, resulting in a higher electron temperature determined by the absorbed energy. However, a coupling to a bosonic bath of phonons can draw out the energy by transferring it to the lattice, which has a larger heat capacity than the electronic system. The determining factor for the dynamics is then the rate at which e–p scattering dissipates the energy, which depends critically on the e–p coupling strength and the phonon frequencies. That the relaxation is ultimately dominated by a strong e–p coupling even in Bi2212 is perhaps surprising even though the electronic relaxation has already been shown to act on timescales much faster than the thermalization timescale observed in tr-ARPES[8,21].

These considerations lead, in their simplest form, to so-called '$n$-temperature' models for relaxation of an electron system driven out of equilibrium[22]. However, the full dynamics of this process, including e–e and e–p interactions, involves a complex interplay of electronic and bosonic interactions beyond this simple model. Energy relaxation of the photoexcited carrier population is expected to occur on three distinct timescales[18]: electron population redistribution on a femtosecond timescale mediated by e–e scattering, transfer of energy from the electrons to strongly coupled phonons within tens to hundreds of femtoseconds and thermalization of the full system of electrons and phonons on picosecond or longer timescales, on which the excited population of the strongly coupled phonons decays through anharmonic coupling or transport. In Fig. 1, we illustrate the first and second processes, which are the relevant ones for the short-term dynamics. The measured and calculated time traces typically show a rise at $t = 0$, followed by a decay. The dynamics are determined by a combination of the first and second processes, where e–p and e–e scattering cooperate in a fashion inconsistent with Matthiessen's rule, $\tau_{\text{total}}^{-1} = \sum_i \tau_i^{-1}$, to equilibrate

the electrons with each other as well as with the phonon bath. Here the breaking of Matthiessen's rule derives from the difference between scattering processes which conserve energy, such as e–e interactions, and those that do no, such as e–p interactions. This difference is what provides our technique's advantage over static ARPES. Further, since only e–p interactions can carry energy away from the electrons in a non-magnetic system, we assign the observed bosonic mode to a phonon (although this is not, strictly speaking, required).

To show that the e–p scattering is the determining factor in the return to equilibrium, we performed simulations of the pump–probe process using non-equilibrium Keldysh Green's functions. We treat the e–p and e–e interactions self-consistently, with both interactions being evaluated at second order in perturbation theory. Here, we go beyond our previous theoretical work[18,23,24] and include the effects of e–e scattering in second order self-consistently. We calculate the time-dependent density $n(E, t)$ along the Brillouin zone diagonal after excitation by an ultrashort 1.5 eV pump pulse for a simple nearest-neighbor tight-binding model and then fit the time traces with a single exponential decay. To illustrate that the resulting decay rates reflect the e–p scattering even when such strong e–e scattering is present, we obtain the decay rates for various e–e scattering strengths $U$. Figure 1b shows the decay rates obtained for quasiparticles at energy $E$ above $E_F$. The rates without e–e scattering show a step at the phonon frequency[18,24]. Strikingly, even when the e–e scattering is increased beyond the e–p scattering strength ($U^2 > 0.02$), the step remains visible in the data. We also note that the high-energy decay rate is smaller when e–e scattering is present, suggesting that there is some competition between the two processes. This originates from the fact that the population dynamics is a fundamentally different quantity than the dynamics (and therefore lifetime) of a singly-excited quasiparticle. The latter is determined by the wavefunction decay, and the former by the interactions of the quasiparticles with each other, by the energy transfer mediated by the phonons and by their final redistribution in phase space.

**Experiment.** We now demonstrate that this analysis can be applied in real materials by making a quantitative comparison between theoretical simulations of the type shown above and a tr-ARPES experiment. A typical equilibrium spectrum $I(E, k_{||}, t_{\text{eq}})$ along the nodal direction of Bi2212 is shown in Fig. 2a. Here $I$ is the photoelectron intensity probed at $t_{\text{eq}} = -5$ ps before arrival of the pump pulse. Although the dispersion kink is difficult to observe without spectral deconvolution at this resolution[25], the increase in coherence (or decrease in linewidth) of the states between $-\hbar\Omega$ and $E_F$ is still clear. In the e–b picture, this is due to a decrease in scattering rate for energies below the boson energy $\hbar\Omega$. Figure 2b shows the difference spectrum from equilibrium, with $t_{\text{eq}}$ representing an equilibrium spectrum:

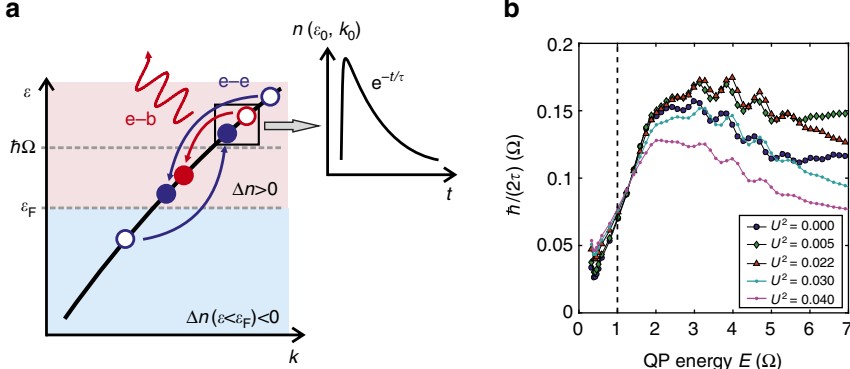

**Figure 1 | Time domain separation of electron–electron and electron–boson scattering.** (**a**) Illustration of the dissimilarity between electron–boson (e–b) and electron–electron (e–e) scattering for population dynamics. Electron–electron scattering (blue arrows) by nature maintains the amount of energy in the electrons, limiting it to rapid redistribution of energy within the electronic subsystem. On the other hand, e–b (red arrows) scattering can carry energy out, leaving a measurable impression on the population dynamics. This causes the population dynamics to decay with the characteristics of the e–b scattering (inset). (**b**) Calculated population decay rate extracted from the time-dependent density $n(E, t)$ along the zone diagonal $k_x = k_y$ as a function of the quasiparticle energy $E$ for various Coulomb interaction strengths $U$. Here, $g^2 = 0.02$ (all interaction strengths are in $eV^2$) and the phonon frequency $\Omega = 0.1$ eV.

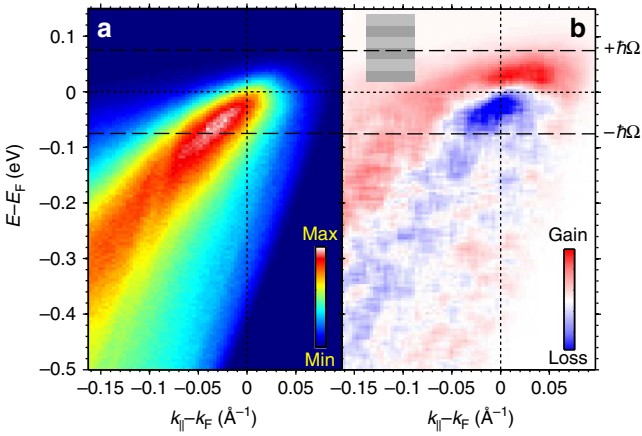

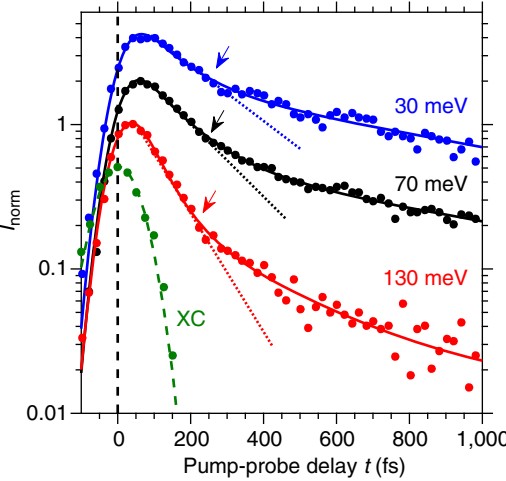

**Figure 2 | Equilibrium and excited photoelectron spectra.** (**a**) Photo-electron intensity $I(E, k_{\parallel}, t_{eq})$ along the node of Bi2212 at equilibrium, $t_{eq} = -5$ ps and equilibrium temperature $T = 120$ K. (**b**) Difference spectrum $\Delta I = I(E, k_{\parallel}, t) - I(E, k_{\parallel}, t_{eq})$ for $\Phi = 35\,\mu\text{J}\cdot\text{cm}^{-2}$, $t = 50$ fs. Light and dark gray bars in **b** depict energy bins $\Delta E = 20$ meV centered at $E_{bin}$ used to evaluate $I_{norm}(E_{bin}, t)$.

**Figure 3 | Time-resolved population decay.** Intensities $I_{norm}(E_{bin}, t)$ at $\Phi = 315\,\mu\text{J}\cdot\text{cm}^{-2}$ for selected binding energies $E_{bin} = 30$ meV (blue circles), 70 meV (black circles) and 130 meV (red circles), respectively, on a logarithmic intensity scale. Solid lines are fits to equation (1). Intensities for each $E_{bin}$ are offset by a factor of two for clarity. Dotted lines are guides to the eye extending through $I_{norm}(E_{bin}, t)$ at times for which population decay is dominated by the scattering mechanism responsible for $\tau_{short}$. Green points are from $I_{norm}(E, t)$ integrated over the $k_{\parallel}$ range shown in Fig. 2 and $1.45 < E < 1.5$ eV for $t < 200$ fs, representing mainly the cross-correlation (XC) time of pump and probe pulses. The Gaussian fit to the green points (green dashed line) yields the XC width. Arrows denote the time at which $\tau_{long}$ dominates $I_{norm}(t)$.

$\Delta I(E, k_{\parallel}, t) = I(E, k_{\parallel}, t) - I(E, k_{\parallel}, t_{eq})$. In Fig. 2b, $t = 50$ fs, and at absorbed pump fluence $\Phi = 35\,\mu\text{J}\cdot\text{cm}^{-2}$. For this fluence, by $t = 50$ fs electrons excited above the equilibrium kink energy $+\hbar\Omega$ and holes injected below $-\hbar\Omega$ have largely relaxed, while those within the boson window $-\hbar\Omega < E < +\hbar\Omega$ persist. Considering that the pump photon energy of 1.5 eV (see Methods section for details) is well in excess of this $2\hbar\Omega$ range, our tracking of the excitations in the vicinity of $E_F$ implies that we are primarily observing secondary electrons and holes and their relaxation towards $E_F$ rather than the initial excitation[17]. Indeed, Fig. 2b shows the largest pump-induced intensity changes within the boson window. The existence of this pileup in excited carriers at finite $t$ means the relaxation times for carriers outside the boson window must be considerably shorter than inside since the former carriers have already relaxed by this time[9,24]. To determine an upper bound of the relaxation time of primary excited electrons $\tau_{ee}$, we plot in Fig. 3 the time-dependent photoemission intensity at $E - E_F = 1.45$–1.5 eV. The observed symmetric time dependence implies that $\tau_{ee}$ is below the effective time resolution of $\sim 10$ fs, which we explain by very efficient e–e interaction, which may involve spin fluctuations as discussed in ref. 21. Therefore, this time-dependent signal serves as a cross-correlation (XC) at the sample of pump and probe pulses and defines $t = 0$. Below, we show that the significant change in relaxation times in the unoccupied part of the spectrum around $+\hbar\Omega$ results from the sudden change in phase space with increasing $E$ for the inelastic decay of excited carriers by emission of a single predominant bosonic mode. This conclusion is based on quantitative agreement between our experiment and the theory of fluence-dependent, boson-derived relaxation times.

To quantify these effects, we introduce a procedure to energy resolve the photoexcited population relaxation times $\tau$ above $E_F$ in a manner that allows comparison with the equilibrium quasiparticle lifetime $\tau_{QP}$ determined by analysis of energy and momentum distribution curves (EDCs and MDCs, respectively) in equilibrium ARPES. This is accomplished by dividing the tr-ARPES spectrum into discrete energy bins of equal width $\Delta E = 20$ meV (Fig. 2b). Integrating the intensity within bins over $k_{\parallel}$ and $\Delta E$ results in time- and energy-dependent intensities $I(E_{bin}, t)$, where $E_{bin}$ is taken at the bin centres.

Our subsequent analysis follows from the observation of a pronounced biexponential decay of photoexcited carrier

populations with respect to $E_{bin}$ and $\Phi$. Typical energy-resolved data for $\Phi = 315\,\mu\text{J}\cdot\text{cm}^{-2}$ is shown in Fig. 3. The normalized intensities $I_{norm} \equiv I(E_{bin}, t)/\max[I(E_{bin}, t)]$ are seen to decay, after an initial rise, with clearly separable short and long time components ($\tau_{short}(E_{bin})$ and $\tau_{long}(E_{bin})$, respectively), the former lasting several hundred femtoseconds. The direct observation of these two exponential timescales within the remaining population decay enables the present analysis of the electron population decay mechanisms in the cuprates in terms of separate electron–optical phonon and electron–acoustic phonon interactions. Fits to $I_{norm}(E_{bin}, t)$ shown in Fig. 3 determine $\tau_{short}$ and $\tau_{long}$ and are performed using the function

$$I_{norm}(t) = \Theta(t)\left(A_{short}e^{-\frac{t}{\tau_{short}}} + A_{long}e^{-\frac{t}{\tau_{long}}}\right) \otimes R_t \quad (1)$$

where $\Theta(t)$ is the Heaviside function, $A$s are intensity amplitudes and $R_t$ is a unit-normalized Gaussian of width equal to the XC of pump and probe responses at the sample surface.

To facilitate examination of both the energy and fluence dependencies of $\tau_{short}$ and its comparison with theory, we show representative $I_{norm}(t)$ for several $E_{bin}$ and $\Phi$ together with the fits in Fig. 4a–c. The main trends in the data are the overall decrease in $\tau_{short}$ with increasing $\Phi$ for $E < \hbar\Omega$, a behavior that reverses outside the boson window (top panels of Fig. 4) and is also obtained in the calculations (bottom panels of Fig. 4).

Plotting $\tau_{short}(E_{bin})$ for each fluence (Fig. 5a), we observe a very significant relationship between the fluence dependence of $\tau_{short}(E_{bin})$ and the value of $E_{bin}$ relative to $\hbar\Omega$. At the lowest fluence, $\tau_{short}(E_{bin})$ shows a pronounced step at $E_{bin} \sim \hbar\Omega$ with much shorter times above $+\hbar\Omega$ versus below. As pump fluence is increased, relaxation times below $\hbar\Omega$ decrease, while those above $\hbar\Omega$ increase. The step in $\tau_{short}$ at $\hbar\Omega$ for $35\,\mu\text{J}\cdot\text{cm}^{-2}$ is reminiscent of the step in $\hbar/\tau_{QP}$ (Fig. 5b), accompanying boson-induced mass renormalizations observed in ARPES as a kink in

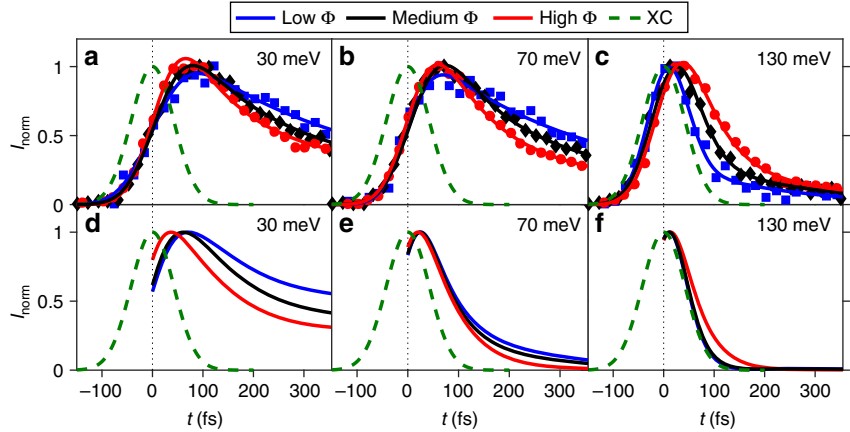

**Figure 4 | Experimental and theoretical fluence-dependent relaxation.** (**a–c**) $I_{norm}$ for $E_{bin} = 30$, 70 and 130 meV, respectively, for $\Phi = 35$ (blue squares), 105 (black diamonds) and 315 $\mu J \cdot cm^{-2}$ (red circles). Solid lines in these panels are fits to equation (1). (**d–f**) Theoretical $I_{norm}$ for the same $E_{bin}$ as **a–c** for field strengths of 0.05 (blue), 0.15 (black) and 0.5 (red) in units of $V/a$, where $a$ is the Cu–Cu lattice constant. (**a–f**) Green dashed lines show the Gaussian cross-correlation (XC).

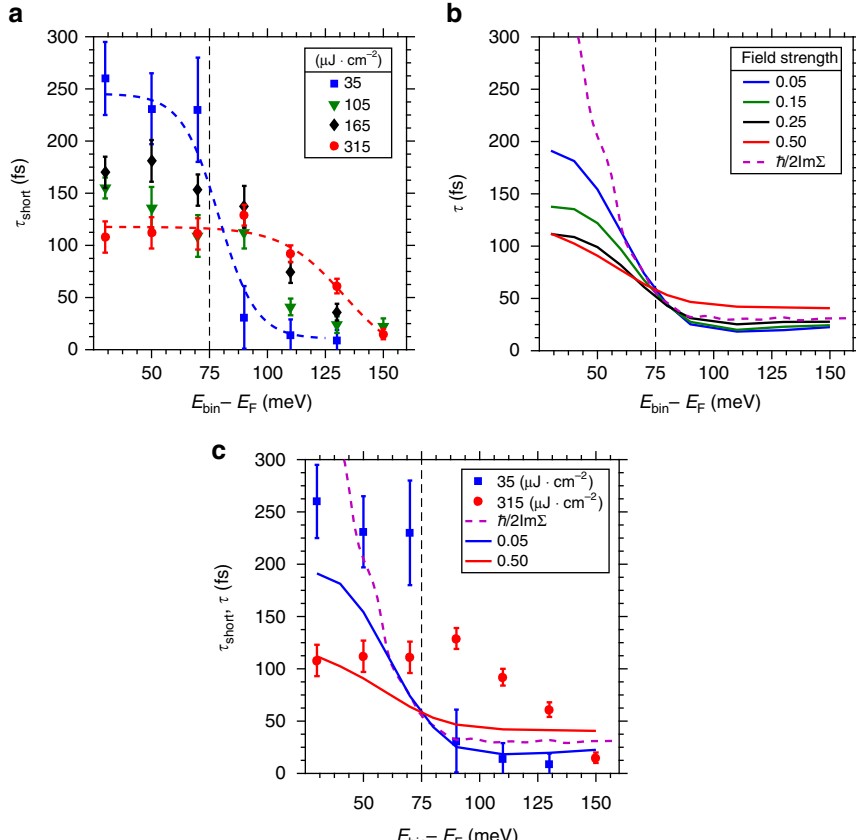

**Figure 5 | Fingerprint of electron–boson scattering.** (**a**) Experimental $\tau_{short}(E_{bin})$ for $\Phi = 35$ (blue squares), 105 (green triangles), 165 (black diamonds) and 315 (red circles) $\mu J \cdot cm^{-2}$, respectively. Dotted blue and red lines are guides to the eye for the 35 and 315 $\mu J \cdot cm^{-2}$ data, respectively. (**b**) Theoretical $\tau_{short}(E_{bin})$. The purple dashed line shows equilibrium $\tau_{QP} = \hbar/2Im\,\Sigma$. (**c**) Direct comparison of highest- and lowest-fluence measurements, respectively, to highest- and lowest-field strength calculations, as well as model $Im\,\Sigma$. The mode energy at 75 meV is marked by the dashed black lines. Error bars on experimental data are statistical, from standard deviation method.

the band dispersion[1,2,5,26]. In ARPES, a single photohole injected above a bosonic mode energy has a shorter lifetime than one injected below because an additional interaction channel is open to the higher energy state. In EDCs this phenomenon is observed as a step, centred at the mode energy, in the Lorentzian widths of the states[20]. Similarly, an electron photoexcited above a bosonic

mode's energy will decay faster than one excited to empty states below because the additional channel for energy loss due to boson emission is open to it, exactly as is observed in experiment and theory in, quantitative agreement[18].

The fluence dependence of these phenomena indicate a potential connection between the step in $\tau_{short}$ observed above

$E_F$ and the nodal kink observed below. It is the process of 'filling' states within the boson window[18] that leads to the characteristic speeding up and slowing down observed here as well as the weakening of the 70 meV kink structure observed in previous experiments[25,27]. As $\Phi$ is increased the dynamics of photoexcited carriers change from a regime reflecting essentially single-particle physics to one which is characterized by the dynamics of populations of interacting, hot particles. The parallel evolution, with increasing fluence, of the broadening step in $\tau_{short}$ and the previously observed reduction in effective mass of the nodal kink below $E_F$ therefore encourages a description of the former effects within the same theoretical framework as the latter. The absence of a similarly pronounced fluence dependence for $\tau_{long}(E_{bin})$ (see Methods) may indicate that it originates in the coupling of hot electrons and phonons to a broad spectrum of bosonic modes, such as acoustic phonon modes, which can efficiently dissipate energy away from the excited system and effectively act like the energy bath in our model. Similarly, at present we can only speculate on the origin of the experimentally observed increase in apparent binding energy of the step in lifetimes as pump fluence is increased (Fig. 5a). It might arise from effects due to electron correlations and changes in phonon properties at higher fluence as well as a hardening of the phonon mode itself under strong fluence. Such a particular fluence dependence of the relaxation times can furthermore indicate saturation of the particles available for excitation and interaction of the excited particles among each other leading to a bottleneck in the relaxation. Such behavior bears the potential to distinguish phonon and spin fluctuations unambiguously[28]. A quantitative analysis will be the topic of a future publication, since such effects are at present not included in our model calculations.

**Comparison of experiment with theory**. With e–p scattering being the dominant mode of energy dissipation, we now focus on a theoretical description of the transfer of energy from the excited electron population to strongly coupled phonons, given experimentally by $\tau_{short}$, and described by the Holstein model, where electrons are coupled to a single Einstein mode. Since the main feature and overall scale is expected to be due to the phonon coupling as discussed above, we neglect e–e scattering here. Excited carriers relax via their coupling to this reservoir, which is assumed to have infinite heat capacity and remains unchanged. The equilibrium $\tau_{QP}(E)$ in Fig. 5b is defined through the equilibrium linewidth due to a single well-defined phonon mode centered at $\hbar\Omega = 75$ meV. The dimensionless e–p coupling constant $\lambda$ is chosen to be 0.2 to match the relaxation times of $\approx 10$ fs at $E > \hbar\Omega$. The same value for $\lambda$ was found account for the coupling of hot electrons to a particular subset of hot phonons in earlier tr-ARPES experiments on Bi2212 (ref. 8), and is similar to that obtained from some equilibrium ARPES experiments[3]. The model also includes a weakly coupled mode at low energies to prevent an infinite $\tau_{QP}$ within the phonon window ($-\hbar\Omega < E < +\hbar\Omega$).

Representative $I_{norm}(E_{bin}, t)$ extracted from theoretically generated time-dependent occupied spectral functions are shown in Fig. 4d–f for field strengths comparable to the experiment. They successfully reproduce the reversal of the fastest $\tau_{short}(E_{bin})$ as a function of fluence for $E_{bin}$ above as compared to below $+\hbar\Omega$. Here, we focus on e–e and electron–optical phonon processes, and ignore the long-time component, fitting the calculated $I_{norm}(E_{bin}, t)$ with a single exponential to produce theoretical $\tau_{short}(E_{bin})$ for several fluences (Fig. 5b). The theoretical $\tau_{short}(E_{bin})$ exhibit the same step at $+\hbar\Omega$ as the experimental data in Fig. 5a. The theory largely reproduces the increase in $\tau_{short}(E_{bin})$ for $E_{bin} > +\hbar\Omega$ and decrease in

$\tau_{short}(E_{bin})$ for $0 < E_{bin} - E_F < +\hbar\Omega$ as a function of increasing fluence. The remarkable quantitative agreement between the measured and calculated $I_{norm}(E_{bin}, t)$ and $\tau_{short}$ within the phonon window (Figs 4 and 5, respectively) demonstrates that scattering due to phonon emission is sufficient to describe the population dynamics measured with tr-ARPES, allowing for the measurement of a single component of the equilibrium self-energy. A direct comparison of the experimental and theoretical $\tau_{short}$ as shown in Fig. 5c highlights the quantitative agreement within the phonon window, in particular at low fluences. It is remarkable, given the stark contrast between the model calculation and experiment, that a phonon-only model can capture the broad strokes of the population dynamics. There are differences, including in particular a shift in the edge, which remain a subject for further study.

## Discussion

The experiment and theoretical calculations both capture the dynamics of energy dissipation through electron–phonon inter- actions and their separation from e–e interactions which keep the energy within the electron system. The fingerprint of the e–p interaction through its particular phase-space restrictions is seen in the data through both energy and fluence dependence of the population decay rates. Understanding the detailed nature of the equilibrium self-energy is thought to be vital to understanding a host of phenomena in strongly correlated materials and beyond, not least those derived from non-Fermi liquid properties that would be expected to show up most strongly in electronic channels. In equilibrium ARPES, it is difficult to disentangle phononic from other contributions to the self-energy, especially when several different interactions are present. The combination of e–e, e–b and electron-impurity scattering—just to name a few—collude and increase the overall linewidth, making it difficult to separate a single contribution in static ARPES. With the techniques demonstrated here, we now have the ability to isolate the phonon contributions directly from the total equilibrium self-energy through measurement of population dynamics with tr-ARPES with high temporal resolution.

This should ultimately allow for a quantitative deconvolution of the various contributions to the single-particle self-energy, insight into which is thought to be vital to understanding the high $T_c$ problem in the cuprates. In particular, in revealing the presence of at least three distinct timescales for population relaxation in Bi2212 our study has provided a new window onto the relative strengths of electronic, optical phonon and acoustic phonon couplings, including a strong bound on the timescale on which the e–e interaction manifests itself in Bi2212. Our analysis has further shown, through the separation of $\tau_{long}$ from $\tau_{short}$, that the energy of electrons driven out of equilibrium is not only dissipated (rather than dispersed) primarily by phonons but also that the 75 meV phonon mode which dominates the e–p interaction above $E_F$ comprises a sufficiently short timescale that it can account for a large portion of the equilibrium self-energy. While this fact in itself is not sufficient to identify the origin of the famous nodal kink in the single-particle spectrum, it is very suggestive of at least some role for this mode since bosons couple to electrons above and below $E_F$ with equal vigor.

Taken together, these observations and their theoretical understanding mean that the dominant form of energy dissipa- tion in Bi2212, at least at optimal doping, is by interaction of electrons with the lattice, a rather surprising finding in a prototypically strongly correlated material. Furthermore, given the energy of the phonon mode responsible for $\tau_{short}$ we can speculate this mode is related to the $\approx 80$ meV bosonic mode observed in the same material in recent momentum-resolved

electron energy loss experiments[29]. However, while this phonon mode clearly dominates the relaxation, our lowest fluence (and so closest to equilibrium) data is best explained quantitatively (Fig. 5c) by an e–p coupling constant of only 0.2 when isolated from other contributions to the self-energy. Thus, while seemingly pronounced when isolated by our non-equilibrium measurements, the interaction with this mode, by itself, does not provide an mechanism for high-temperature superconductivity in optimally doped Bi2212, although, as with the kink, its prominence may imply its participation in conjunction with other effects such as those of spin fluctuations. If this phonon mode is a c-axis mode, as seems likely, its ready separation from other interactions may further indicate that it constitutes a uniquely important channel for electronic interaction between the copper oxide planes even for the nodal region, which is typically considered to be highly two-dimensional.

Here, we have focused on a material where the presence of a complex interplay of e–e and e–b interactions is well known, but ultimately separable in the time domain. The very existence of the boson window effect in our non-equilibrium data indicates that for at least several hundred femtoseconds after the excitation of a so-called $n$-temperature model, in which a thermalized electron distribution can be considered to be well described by, for example, a Fermi–Dirac distribution, is clearly invalid because the dynamic spectral weight transfer related to the boson window does not follow such a distribution. This is well past the time that e–e interactions dominate and are often assumed to have thermalized the excited electrons. Further, our high fluence data indicate the potential importance of non-equilibrium effects on the boson population itself and such effects further complicate any effort to describe the relaxation of the system in simple thermodynamic terms. Considering the wide employment of such models in understanding time-resolved experiments in strongly correlated materials, these findings also have important ramifications beyond the present study. They mean that strong correlations are no guarantee as a temperature model will be accurate even to relatively long times from the initial excitation. Theoretical techniques of the sort deployed here are therefore vital for an understanding of the non-equilibrium physics of strongly correlated materials such as the Fe-based superconductors, heavy fermion compounds, and manganites, as well as more weakly correlated materials with strong e–p interactions. Topological insulators are a field where the study of population dynamics is commonly used, and the methodology presented here can aid in the interpretation and analysis of the results[30–36]. Moreover, the particular relevance of bosonic mode couplings for excited state population dynamics is of high interest, for instance, in light-harvesting applications. We point out that this relevance is often implicitly assumed, for example, in recent *ab initio* computations of equilibrium self-energies, from which information about hot carrier dynamics for solar cells was inferred[37]. Our study provides a firm ground for such implicit assumptions, and the methods laid out by us can guide further efforts in this direction.

## Methods

**Sample preparation and set up of tr-ARPES experiment.** Our trAPRES experiment was performed on single crystals of optimally doped Bi2212 ($T_c = 91$ K) grown by the floating zone method. $T_c$ was confirmed by SQUID magnetometry. Samples were cleaved *in situ* at the lattice equilibrium temperature, 100 K, in a base vacuum of $5 \times 10^{-11}$ mbar. Pump pulses of 800 nm wavelength and 40 fs duration, at 250 kHz repetition rate, were produced by a Coherent RegA 9040 regenerative Ti:Sapphire amplifier. The 200 nm probe beam was produced as the fourth-harmonic of part of the RegA's 800 nm fundamental using nonlinear crystals[8]. The pump–probe XC (Gaussian full-width at half-maximum) was determined to be 100 fs at the sample surface. Photoelectron spectra with an overall energy resolution of 55 meV, set by the bandwidth of the laser pulses, were recorded using both angle integrating time-of-flight (TOF)[8] and position-sensitive TOF (pTOF)[38]

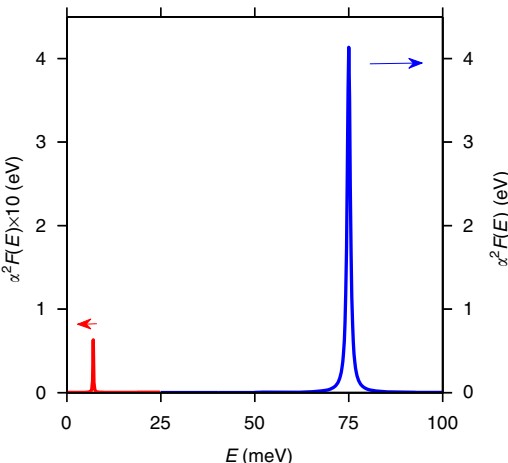

**Figure 6 | Bosonic modes of the model.** (Blue, right axis) Theoretical $\alpha^2 F(E)$ with e–b coupling constant $\alpha$ and dispersion $F(E)$. (Red, left axis) The low energy lifetime dampening mode at 15 meV is magnified by $\times 10$ for clarity.

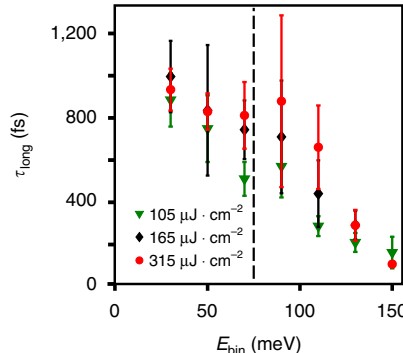

**Figure 7 | Experimental fluence and energy dependence of $\tau_{long}$.** $\tau_{long}$ versus $E_{bin}$, as defined in the main text, for $\Phi = 105$ (green) 165 (black) and 315 (red) $\mu J \cdot cm^{-2}$. The vertical dashed line at 75 meV denotes the theoretical position of $+\hbar\Omega$.

electron spectrometers. The $k$ resolution of the pTOF was 0.0025 Å$^{-1}$ and the angular acceptance of the TOF was $\pm 3°$.

**Theoretical modelling of the data.** For the modelling, we use a tight binding parametrization of the one-band model for cuprates[39] with a filling of 0.42 per spin, that is, 16% hole doping. The methodology for the calculations is a self-consistent Keldysh Green's function approach as described in ref. 18.

We couple the electrons in this band to a spectrum of bosons characterized by an Eliashberg function $\alpha^2 F(E)$ for a spectrum of modes with frequency $\Omega_\gamma$, which is

$$\alpha^2 F(E) = \sum_\gamma g_\gamma^2 \delta(E - \hbar\Omega_\gamma). \qquad (2)$$

In particular, we use a dominant boson mode centered at $\hbar\Omega = 0.075$ eV, with $g^2 = 0.0065$ eV$^2$ corresponding to dimensionless coupling $\lambda = 0.2$, extracted from the slope of the real part of the retarded equilibrium self-energy at zero energy:

$$\lambda \equiv -\left.\frac{\partial \mathrm{Re}\Sigma(E)}{\partial E}\right|_{E=0}. \qquad (3)$$

To ensure finite relaxation times at low energy, we add a second weakly coupled mode centered at 0.007 eV with $g^2 = 2 \times 10^{-5}$ eV$^2$. The frequency of the strongly coupled bosonic mode was chosen to match the experimentally determined position of the crossing point of fluence- and energy-dependent decay times, as well as the nodal kink position. The coupling strength was adjusted to give roughly the correct timescale in the low-fluence limit for $E > \hbar\Omega$. $\alpha^2 F(E)$ for both modes is shown in Fig. 6.

We excite this metallic single-band model system by a spatially homogeneous pump pulse, whose vector potential couples to the band electrons through the Peierls substitution $\mathbf{k} \rightarrow \mathbf{k} - e\mathbf{A}(t)$, where $\mathbf{k}$ is wavevector, $e$ is the electron charge and $\mathbf{A}(t)$ is the time-dependent vector potential. (Here the speed of light $c = 1$.) The vector potential $\mathbf{A}(t)$, which is in the Hamiltonian gauge, points along the zone diagonal and has a temporal shape with a Gaussian envelope ($\sigma = 20$ fs), oscillation frequency $\hbar\Omega = 0.35$ eV and peak strength $A_{max}$. For the simulations in this work, we use field strengths $A_{max} = 0.05$, $0.15$, $0.25$ and $0.50$ in units of $V/a_0$. For the given parameters and a typical lattice constant $a_0 = 3.8$ Å (Cu–Cu distance), these field strengths correspond to peak electric fields of 0.65, 2.0, 3.3 and 6.5 MV cm$^{-1}$, respectively. The tr-ARPES signal is obtained in a postprocessing step using the formalism described in refs 24,40 with probe pulse of width $\sigma_{pr} = 40$ fs. The energy-resolved intensity is obtained by integrating the momentum- and energy-resolved data along the Brillouin zone diagonal cut. The decay times shown in the main text (Fig. 4d–f) are extracted from fits to the normalized intensity changes, binned in energy windows of 20 meV width like the experimental data and convolved with a Gaussian envelope function of full-width XC to render the theoretical results directly comparable to the experiment. The present modelling and analysis are based mostly on the energy scale (70–75 meV), extracted from the nodal kink position (below $E_F$) as well as the reported population decay time step position (above $E_F$). Importantly, statements about coupling to bosonic modes with energies higher than 100 meV cannot be drawn from our analysis.

**Theoretical comparison of e–e and e–p scattering.** The e–e interactions are treated self-consistently at second order with a local interaction strength $U_0$:

$$\Sigma^C(t, t') = -i^2 U_0^2 G^C(t, t') G^C(t, t') G^C(t', t) \tag{4}$$

where $G^C(t, t')$ is the double-time contour-ordered Green's function[41]. Since the interaction strength depends on the filling, we define an effective $U : U \equiv U_0^2 n(1 - n)$ for a better comparison to the e–p coupling strength. For the comparison, we used a simplified tight-binding model with $t = 0.25$ eV, $t' = 0.075$ eV and $\mu = -0.255$ eV. We use a strongly coupled phonon with $\Omega = 0.1$ eV and $g^2 = 0.02$ eV$^2$. It should be noted that the parameters for the comparison were chosen to make the calculations computationally feasible and make a clear distinction between e–e and e–p scattering, rather than for a quantitative comparison with the data (as in the other sections of the manuscript).

**$\tau_{long}$ versus $E_{bin}$ and $\Phi$.** The long timescale $\tau_{long}$, shown in Fig. 7, was found to carry no fluence dependence, within error. Interestingly, it was found to nearly merge with or at least become indistinguishable, from $\tau_{short}$ at high energies, although higher statistics and time resolutions would be required to ascertain this with certainty. Regardless, $\tau_{short}$ is found robustly at all fluences measured. The fluence independence of $\tau_{long}$ have an important role in our extraction of $\tau_{short}$ at the lowest fluence, particularly around $E_{bin} = 70$ meV. Because at $\Phi = 35 \mu J \cdot cm^{-2}$ the statistics are relatively low as $E_{bin}$ increases much past 70 meV, and at later times, consistent fitting requires $\tau_{long}$ to be held constant from higher fluence data. Relaxing this constraint of course does not alter the trends for $\tau_{short}$ visible in the raw data, nor does it appreciably alter the results for the lowest fluence in, for example, Fig. 4a of the main text viz a viz the step around 70 meV. It does introduce a bit more uncertainty into the error bars but the position of the step remains.

**Data availability.** The data that support the findings of this study are available from J.D.R., U.B. and A.F.K. on reasonable request (jrameau@bnl.gov) (uwe.bovensiepen@uni-due.de) (akemper@ncsu.edu).

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

## Acknowledgements

Work at Brookhaven National Laboratory was supported by the Center for Emergent Superconductivity, an Energy Frontier Research Center, headquartered at Brookhaven National Laboratory and funded by the US Department of Energy, under Contract No. DE-2009-BNL-PM015. This work was supported, in part. by National Science Foundation Grant No. PHYS-1066293 and the hospitality of the Aspen Center for Physics. A.F.K. was supported by the Laboratory Directed Research and Development Program of Lawrence Berkeley National Laboratory under US Department of Energy Contract No. DE-AC02-05CH11231. J.K.F. was supported by the Department of Energy, Office of Basic Energy Sciences, Division of Materials Sciences and Engineering (DMSE) under Contract No. DE-FG02-08ER46542, and by the McDevitt bequest at Georgetown. M.A.S. received further support from the Deutsche Forschungsgemeinschaft (DFG) through the Emmy Noether program. Computational resources were provided by the National Energy Research Scientific Computing Center supported by the Department of Energy, Office of Science, under Contract No. DE-AC02-05CH11231. We acknowledge further funding from the Deutsche Forschungsgemeinschaft through SFB 616 and SPP 1458, from the Mercator Research Center Ruhr through Grant No. PR-2011-0003 and from the European Union within the seventh Framework Program under Grant No. 280555 (GO FAST).

## Author contributions

Photoemission data for the experiments was taken by J.D.R., S.F., I.A., M.L., U.B., P.D.J. and L.R. and analyzed by J.D.R., S.F. and U.B. Samples were grown and characterized by Y.Y., H.E., R.D.Z., Z.J.X. and G.D.G. Theoretical modeling and calculations were performed by A.F.K. and M.A.S. The manuscript and figures were prepared by J.D.R., A.F.K., M.A.S., J.K.F., M.L., P.D.J. and U.B. and commented on by all authors.

## Additional information

**Competing financial interests:** The authors declare no competing financial interests.

**Publisher's note**: 

