## [Peer Review File · Nature Communications]

Reviewers' comments:

Reviewer #1 (Remarks to the Author):

The manuscript by Rameau et al provides a combined theoretical and experimental analysis of the out of equilibrium electronic structure of BSSCO - a prototype system for ARPES studies of high temperature superconductors - following excitation with femtosecond light pulses.

Thanks to this combined effort, the manuscript provides a nice theoretical framework for ultrafast studies of complex materials. These studies are normally treated with approaches based mainly on reasonable assumptions and physical intuition. In this sense, the paper is original and interesting, and a very welcome and much needed initiative in this research area.

The approach is valid, based on well established experimental data, and the presentation is clear. The conclusions based on the theoretical analysis are almost certainly valid.

Nevertheless, it is a bit disappointing that no genuinely novel information on the physics of this model material could be extracted from this joint effort. The paper essentially confirms the validity of the conclusions of previous "semi-empirical" approaches, but doesn't open new perspectives in our understanding of out of equilibrium BSSCO.

The authors may want to try to develop some points, to extract some novel physical information, or at least to make it more explicit to the reader.

For instance, the pump-probe ARPES results of ref. 7 indicated a weak (average) electron-phonon coupling in BSSCO, while ARPES analysis (for ex. refs. 1, 19) supports a strong coupling to the lattice. Reading the manuscript, the authors use for the dimensionless coupling a value of $\lambda=0.2$ (page 15), which is rather weak; on page 16, they say that their theoretical comparison is based on a "strongly coupled phonon". Based on their joint effort, can the authors make a firm statement on the strength of the e-ph coupling in this system? if it is selective, can it contribute to the mechanism of superconductivity?

Another point that might be interesting to develop is how electron-electron correlations are related to the typical thermalisation time for the system, i.e. the time necessary to be able to describe the electrons with a Fermi-Dirac distribution with a "hot" T , compared to the early stages after photoexcitation where a more Maxwell-Boltzmann-like distribution is dictated. This time is very short in BSSCO: if the authors reduce e-e correlation, can they reproduce the markedly slower thermalisation behaviour of less correlated systems, like Dirac electrons in topological insulators (refs. 28-33)? or (related question), can the authors give indications on when one can safely adopt a "two-temperature" or "three-temperature" model to analyze experimental data?

As a minor detail, in the caption of figure 1, I suppose the phrase "This causes the population dynamics to decay..." refers to the inset between figures a and b: one should probably explicitly mention that for the general reader.

Overall, I believe that the authors should try to convey some novel piece of information from their effort. Otherwise, in my opinion the manuscript falls short of meeting the criteria expected for Nature Communications.

Reviewer #2 (Remarks to the Author):

The manuscript presents an experimental and theoretical study of tr-ARPES. The main goal of the paper appears to be to show that population dynamics can be a more effective way in disentangling the role of el-el and el-ph interaction strength in complex materials. This is because the dynamics in tr-ARPES is strongly controlled by phase space available for inelastic scattering, and this can be very different when the inelastic scattering is due to electrons vs phonons. To demonstrate this, the focus of the work is to show how a particular optical phonon mode in Bi accounts for the short time dynamics by giving very fast relaxation for energies $>$ phonon frequency, and slowing down the dynamics for energies $<$ phonon frequency.

While the main goal of the paper is a nice one, I have the following questions:

Q1: Fig 4 and Fig 5. The short time rates obtained from experiment and theory should be plotted together to see to what extent the rates obtained experimentally are a good measure for electron-phonon scattering time. Right now the plots only show that relaxation speeds up or slows down depending on the presence or absence of phonon modes.

However, can the experiment be used to extract el-ph self-energies quantitatively? If not, what exactly can these kind of experiments hope to give us? Besides the already known fact that some particular phonon modes are active/inactive in some particular energy windows, is there more information to extract from these measurements?

Q2: Both the theory and experiment show a strange behavior where when the fluence increases, the relaxation rate slows down at some energies. Why does this happen? I see that even the theory shows this to some extent, so some explanation is warranted.

Q3: The discussion on Page 10 second para: "The fluence dependence of these phenomena indicate a strong connection between and nodal kink". And again later in the paragraph "previously observed reduction in effective mass of the nodal kink below E_f" This discussion is not at all clear. The location of the nodal kink below E_f in equilibrium ARPES has to do with coupling to a boson mode (for this case the mode at approximately 70 meV). What does this nodal-kink and reduction of effective mass have to do with the fluence dependence observed in the experiment?

Please explain clearly.

After all these questions have been answered, I can make my recommendation.

REVIEWERS' COMMENTS:

Reviewer #1 (Remarks to the Author):

I find this new version of the manuscript considerably improved. The Authors have made a definite effort to clarify some weak points and to explain the possible impact of their results. It is a very interesting paper, valuable for scientists working in this field and accessible to a broad readership.

I do recommend its publication in Nature Communications.

Reviewer #2 (Remarks to the Author):

The revised manuscript, together with the response to the two referees explains much more clearly the new aspects of this work.

Overall it is a good quantitative comparison between theory and experiment that in the process also explains the role of the optical phonon mode, and its importance relative to electron-electron interactions on relaxation dynamics. I therefore recommend it be published in Nature Communications.

NCOMMS-16-07328A

Note: reference numbers in our replies have been changed to reflect the order in the revised manuscripts.

Reviewer #1 (Remarks to the Author):

The manuscript by Rameau et al provides a combined theoretical and experimental analysis of the out of equilibrium electronic structure of BSSCO - a prototype system for ARPES studies of high temperature superconductors - following excitation with femtosecond light pulses.

Thanks to this combined effort, the manuscript provides a nice theoretical framework for ultrafast studies of complex materials. These studies are normally treated with approaches based mainly on reasonable assumptions and physical intuition. In this sense, the paper is original and interesting, and a very welcome and much needed initiative in this research area.

The approach is valid, based on well-established experimental data, and the presentation is clear. The conclusions based on the theoretical analysis are almost certainly valid.

We thank the referee for his/her careful reading of the manuscript and the positive commentary.

Nevertheless, it is a bit disappointing that no genuinely novel information on the physics of this model material could be extracted from this joint effort. The paper essentially confirms the validity of the conclusions of previous "semi-empirical" approaches, but doesn't open new perspectives in our understanding of out of equilibrium BSSCO.

We respectfully disagree with the referee that no novel information is obtained. First, we wish to point out (as the referee also noted) that the point of the manuscript is to demonstrate that there is a fundamental difference between measurements made in- and out of equilibrium. As soon as more than one type of scattering is present (e.g. electron-electron and electron-phonon, as we consider here), the population decay rate and equilibrium scattering rate reflect different quantities.

Next, our experiment demonstrates the existence of, at minimum, *three* distinct timescales for the relaxation of excited electrons in BSCCO: 1) a fast decay time not exceeding the ~50 fs half-width of the pump-probe cross correlation with no discernible energy dependence, 2) an intermediate timescale ranging from a few hundred fs below a threshold energy of ~80 meV above E_F and tens of fs above this and a pronounced pump-fluence dependence and 3) a long timescale of many hundreds of fs with a linear energy dependence independent of pump fluence. The quantification of these timescales is made possible by the enhanced temporal resolution and photoelectron sensitivity (due to the inherent "pulse counting" nature of the experimental setup) relative to previous trARPES experiments performed on similar samples [Refs 9,17]. In particular, the identification of the fastest timescale in trARPES is entirely new, and the teasing apart of the second and third timescale a significant advance over earlier measurements. The separate identification of these timescales in trARPES represents a significant advancement of our understanding of the out-of-equilibrium properties of BSCCO, particularly in the normal state.

Theoretical calculations, in conjunction with basic observations from the data serve to elucidate the origins of these timescales.

The shortest timescale (≤ 50 fs), reflects the fast relaxation of carriers excited into the charge-transfer gap at all momenta and all energies up to the 1.5 eV pump photon energy [green curve, fig 3]. Since these mid-gap excitations are subject to the double occupancy restriction of the underlying Cu lattice, their decay is likely to proceed by some form of e-e interaction. Indeed recent optics experiments have shown the dominant relaxation mechanism within this time to arise from electronic (spin fluctuation) interactions [S. Dal Conte et al., Science 30, Vol. 335, Issus 6076, pp. 1600 (212)]. Presently, we observe and explain by explicit calculation that these interactions only rearrange the carriers population but to not lead to any loss energy from the excitation. This is most prominently evidenced by the fact that most of the spectral weight changes we observe occur within ~ 70 meV of E_F and not up to the 1.5 eV one might expect from the pump photon energy alone.

In fact, these changes occur primarily within the boson window set by well-known optical phonon energies in the cuprates, and on timescales concomitant with what one expects for relaxation by electron-lattice interactions both from first principles and from the explicit calculations we present. The experimental separation of this timescale from the longest timescale greatly increases the certainty of this identification for the primary high-energy decay mechanism of out of equilibrium electrons in BSCCO (outside the boson window). This slowest relaxation on the other hand is due to the presence of an immutable thermal "bath" of low energy bosonic degrees of freedom which appear to couple primarily to carriers within the boson window, thus becoming observable only when the previous two mechanisms have relaxed the excited distribution.

The population dynamics, as measured with a time-resolved probe, show temporal behavior that is controlled principally by the dynamics of energy transfer between the various subsystems, which means the interactions present in the system show up differently in this situation. This opens an entirely new perspective on many materials, and certainly on BSCCO. We illustrate that, using this methodology, we can make a quantitative statement regarding the strength of the electron-boson coupling in this material, without obfuscation by dopant (impurity) scattering and possibly even electron-electron interactions.

Finally, we disagree in a sense with the previous semi-empirical approaches. These tend to follow the lines of a Boltzmann approach, either through an effective temperature or effective distribution description. In said approaches, all types of interactions are treated on an equal footing. In this work, we have demonstrated that for population dynamics, electron-electron and electron-phonon interactions behave differently, and can even counteract. This directly contradicts any approach that simply sums these interactions.

We recognize that our original manuscript was perhaps not as clear as it should have been in demarcating, in particular, this hierarchy of timescales and elucidating how and why these findings are novel or how the experimental findings, when paired with our theoretical analysis, yields new quantitative information about the physics of Bi2212 that could not be obtained with traditional equilibrium methodologies. We have therefore revised the introduction to more explicitly explain this and the discussion to elucidate what impact these findings have on our understanding of, in particular, Bi2212, especially as regards the surprising role phonons play in dissipating energy from the system when driven out of equilibrium.

The authors may want to try to develop some points, to extract some novel physical information, or at least to make it more explicit to the reader.

For instance, the pump-probe ARPES results of ref. 7 indicated a weak (average) electron-phonon coupling in BSSCO, while ARPES analysis (for ex. refs. 1, 19) supports a strong coupling to the lattice. Reading the manuscript, the authors use for the dimensionless coupling a value of $\lambda=0.2$ (page 15), which is rather weak; on page 16, they say that their theoretical comparison is based on a "strongly coupled phonon". Based on their joint effort, can the authors make a firm statement on the strength of the e-ph coupling in this system? if it is selective, can it contribute to the mechanism of superconductivity?

The extraction of electron-phonon coupling constants using static ARPES is difficult, and relies on a number of approximations and assumptions. For one thing, the coupling is generally extracted from either the real or imaginary parts of the self-energy, which inherently contains all interactions. Furthermore, the extraction of the self-energy requires a priori knowledge of the un-renormalized band structure, which is not an independent measurement that can be made using static ARPES. In fact, this has led to a disagreement of the electron-phonon coupling strength, as the referee correctly noted. In our work, based on the fact that the population dynamics are mainly determined by energy transfer, we find that a dimensionless coupling constant of $\lambda=0.2$ reasonably reproduces the experimental results. This is in agreement with some of the ARPES experiments (e.g. refs 3 and 4).

We appreciate that $\lambda=0.2$ may not be seen as a strongly coupled phonon depending on the context, and we have adjusted the language in the manuscript accordingly.

Based on the obtained strength of the electron-phonon coupling, we feel that its contribution to the mechanism of superconductivity is at most a moderate enhancement of a spin-mediated mechanism, rather than being the main source. On the other hand, the possibility that the observed phonon is an unusually well-isolated c-axis mode indicates a possibly heretofore unrecognized interaction is at play in these otherwise very two-dimensional materials.

Another point that might be interesting to develop is how electron-electron correlations are related to the typical thermalisation time for the system, i.e. the time necessary to be able to describe the electrons with a Fermi-Dirac distribution with a "hot" T , compared to the early stages after photoexcitation where a more Maxwell-Boltzmann-like distribution is dictated. This time is very short in BSSCO: if the authors reduce e-e correlation, can they reproduce the markedly slower thermalisation behaviour of less correlated systems, like Dirac electrons in topological insulators (refs. 28-33)? or (related question), can the authors give indications on when one can safely adopt a "two-temperature" or "three-temperature" model to analyze experimental data?

We feel that the reduction of complex dynamics to an N-temperature model to analyze experimental data is an approach which, while feasible, oversimplifies the situation. To be able to make this reduction, it is necessary that the electron-electron interactions act sufficiently fast to thermalize the electrons in some sense, before significant energy transfer to the bosons is occurring. Usually, these processes happen contemporarily, and an interplay between the two results, as illustrated in Fig. 1. In fact, there is a non-monotonic dependence of the relaxation rates on the electron-electron interaction strength (in particular at energies above the boson energy). Earliest timescale reflects this phenomenon and clearly precedes the exponential, quasi-thermal decay of a “thermalized” population (2nd and 3rd timescales). We have amended the introduction and discussion sections to clarify these points.

The markedly slowed thermalisation behavior of Dirac electrons in topological insulators as well as graphene is more likely due to phase space restrictions, as discussed by Zhang et al (DOI:[10.1140/epjst/e2013-01920-2](https://doi.org/10.1140/epjst/e2013-01920-2)).

Finally, as we have recently demonstrated ([arXiv:1605.00121](https://arxiv.org/abs/1605.00121) and *Entropy* 2016, 18(5), 180), once the system can be described by a distribution function multiplied by a spectral function, no relaxation can occur unless there exists some coupling to an external bath. This implies that multi-temperature models should be applied with some caution, and we try to avoid doing so.

This topic is now addressed explicitly in the discussion section of our paper, the introduction and the theory section of the results section.

As a minor detail, in the caption of figure 1, I suppose the phrase "This causes the population dynamics to decay..." refers to the inset between figures a and b: one should probably explicitly mention that for the general reader.

We have clarified the discussion in the manuscript.

Overall, I believe that the authors should try to convey some novel piece of information from their effort. Otherwise, in my opinion the manuscript falls short of meeting the criteria expected for Nature Communications.

We have addressed these concerns by considerable addition to the discussion, particularly as regards, as discussed above, N-temperature models' shortcomings, our resolution of three distinct timescales in the trARPES relaxation process and a quantitative determination through comparison to theoretical models of the e-p coupling constant to a particular phonon mode.

Reviewer #2 (Remarks to the Author):

The manuscript presents an experimental and theoretical study of tr-ARPES. The main goal of the paper appears to be to show that population dynamics can be a more effective way in disentangling the role of el-el and el-ph interaction strength in complex materials. This is because the dynamics in tr-ARPES is strongly controlled by phase space available for inelastic scattering, and this can be very different when the inelastic scattering is due to electrons vs phonons. To demonstrate this, the focus of the work is to show how a particular optical phonon mode in Bi accounts for the short time dynamics by giving very fast relaxation for energies $>$ phonon frequency, and slowing down the dynamics for energies $<$ phonon frequency.

While the main goal of the paper is a nice one, I have the following questions:

Q1: Fig 4 and Fig 5. The short time rates obtained from experiment and theory should be plotted together to see to what extent the rates obtained experimentally are a good measure for electron-phonon scattering time. Right now the plots only show that relaxation speeds up or slows down depending on the presence or absence of phonon modes.

However, can the experiment be used to extract el-ph self-energies quantitatively? If not, what exactly can these kind of experiments hope to give us? Besides the already known fact that some particular phonon modes are active/inactive in some particular energy windows, is there more information to extract from these measurements?

In the manuscript, we have aimed to demonstrate that the population decay rates are determined in large part by the electron-phonon interaction. In this material, where electron-electron scattering is known to play an important role, this is quite an unexpected result! As such, these types of experiments allow for the quantitative study of just the electron-phonon scattering part of the self-energy, which is not possible with equilibrium experiments where all interactions occur on the same footing. We have illustrated this by making a quantitative comparison to a theoretical calculation which neglects electron-electron scattering altogether. Because the excitation in experiment is somewhat different from theory, and because the population decay rates are mainly, but not entirely, determined by electron-phonon scattering, the theoretical and experimental curves are similar but not identical, and we have thus plotted them separately. We would like to point out that the absolute scales on the plots are identical, and thus we compare the results quantitatively.

Furthermore, when combined with self-energies measured using static ARPES, the quantitative information on e-ph scattering in trARPES may allow the deconvolution of the respective contributions to the equilibrium self-energies.

Q2: Both the theory and experiment show a strange behavior where when the fluence increases, the relaxation rate slows down at some energies. Why does this happen? I see that even the theory shows this to some extent, so some explanation is warranted.

The relaxation rate slows down when the fluence is increased for quasiparticles at energies above the boson window, i.e. above ~ 75 meV. This occurs due to the process described in ref. 23(DOI:10.1103/PhysRevB.90.075126): The decay rates are in part determined by the availability of phase space for the quasiparticles to scatter into. At low temperatures and low fluence, quasiparticles above the Fermi level and above the boson window are scattering into essentially entirely empty final states. As the fluence is increased, these states become partially occupied by the pumped electrons, limiting the phase space for scattering and thus decreasing the relaxation rate. Below, we have reproduced an illustration of this process from ref. 23 (DOI:10.1103/PhysRevB.90.075126).

Q3: The discussion on Page 10 second para: "The fluence dependence of these phenomena indicate a strong connection between and nodal kink". And again later in the paragraph "previously observed reduction in effective mass of the nodal kink below E_f" This discussion is not at all clear. The location of the nodal kink below E_f in equilibrium ARPES has to do with coupling to a boson mode (for this case the mode at approximately 70 meV). What does this nodal-kink and reduction of effective mass have to do with the fluence dependence observed in the experiment? Please explain clearly.

This relation is also discussed in Ref 23 (DOI:10.1103/PhysRevB.90.075126), and we will outline the salient points here. The effective mass/nodal kink arise from the real part of the electron-boson self-energy ($\text{Re } \Sigma$). These are intimately related to the quasiparticle lifetime, which arises from the imaginary part of the same ($\text{Im } \Sigma$), through the Kramers-Kronig relations. In equilibrium, there is a sharp step in ($\text{Im } \Sigma$), which is due to the phase space restrictions for scattering mentioned above, and a sharp peak in $\text{Re } \Sigma$. When the system is pumped, as also discussed in the previous point, this sharp step in the decay rates due to the phase space restrictions gets smoothed out. Because of the Kramers-Kronig relations, $\text{Re } \Sigma$ – and thus the kink in the dispersion – also smoothen out.

As we now say more clearly in the discussion, the relationship between the mode observed above E_F in the trARPES data and the kink observed below E_F in static ARPES remains tenuous. It's true that e-ph coupling is particle-hole symmetric and so a mode observed to couple strongly above E_F should do likewise below. It's also true that the mode energies are nearly identical and that a kink below E_F is required to reproduce the trARPES results theoretically. However, because other effects are likely at play, such as coupling of electrons to a background of spin fluctuations (Ref. Johnson), we cannot definitively say that the one directly implies the other. Thus, we have qualified the original statement in our revised manuscript.

After all these questions have been answered, I can make my recommendation.

Major Changes to the Manuscript

1. Besides some reordering, three additional references have been added: [21] S.D. Conte et al., [22] P. Allen, [28] J. Kogoj et al. and [29] S. Vig et al.
2. At the request of referee 2 we have added a third panel (panel C) to Figure 5 showing a direct comparison of theoretical and experimental decay times τ for the lowest and highest fluences in the experiment from panel A of the figure and the corresponding theoretical curves from panel B. The caption has been amended to reflect this change.
3. The title is changed to: Energy Dissipation from a Correlated System Driven Out of Equilibrium
4. The introduction has been changed to reflect our discussion of three rather than two characteristic timescales, versus the single timescale discussed in earlier work. In the earlier version, we had initially described three time scales but had not clarified the role or presence of the e-e timescale or discussed the presence of three rather than two scales in the remainder of the text.
5. We further clarify in the introduction why the boson discussed is a phonon.
6. In the Results: Theory section we further elucidate the presence of three rather than two timescales and clarify the role of correlations in the e-e timescale.
7. Last paragraph before "Comparison of Experiment and Theory", paragraph added to provide further insight into shift of the apparent phonon mode energy at higher fluences, beyond what is captured by the theory.
8. Last paragraph before discussion: comment on new Fig 5c and discussion of difference between theory and experiment.
9. The discussion is greatly expanded here to address the remaining concerns of the referees, viz a viz the novelty of the work, its particular relevance to the physics of Bi2212, our ability to extract new, quantitative information from this type of experiment and the applicability, or lack thereof, of so-called "N-temperature" models.